# Spatiotemporal Variations and Risk Analysis of Chinese Typhoon Disasters

**Fang Chen** [1,2,3,4], **Huicong Jia** [1,*] , **Enyu Du** [2], **Lei Wang** [4], **Ning Wang** [1,2] and **Aqiang Yang** [1]

1 Key Laboratory of Digital Earth Science, Aerospace Information Research Institute, Chinese Academy of Sciences, Beijing 100094, China; chenfang_group@radi.ac.cn (F.C.); wangning171@mails.ucas.ac.cn (N.W.); yangaq@radi.ac.cn (A.Y.)
2 College of Resources and Environment, University of Chinese Academy of Sciences, Beijing 100049, China; duenyu@cug.edu.cn
3 Hainan Key Laboratory of Earth Observation, Institute of Remote Sensing and Digital Earth, Chinese Academy of Sciences, Sanya 572029, China
4 State Key Laboratory of Remote Sensing Science, Aerospace Information Research Institute, Chinese Academy of Sciences, Beijing 100094, China; wanglei@radi.ac.cn
\* Correspondence: jiahc@radi.ac.cn; Tel.: +86-10-8217-8135

**Abstract:** Typhoons are a product of air-sea interaction, which are often accompanied by high winds, heavy rains, and storm surges. It is significant to master the characteristics and pattern of typhoon activity for typhoon warning and disaster prevention and mitigation. We used the Kernel Density Estimation (KDE) index as the hazard index; the probability of exceeding, or reaching, return period or exceeding a certain threshold was used to describe the probability of hazard occurrence. The results show that the overall spatial distribution of typhoon hazards conforms to a northeast-southwest zonal distribution, decreasing from the southeast coast to the northwest. Across the six typical provinces of China assessed here, data show that Hainan possesses the highest hazard risk. Hazard index is relatively high, mainly distributed between 0.005 and 0.015, while the probability of exceeding a hazard index greater than 0.015 is 0.15. In light of the four risk levels assessed here, the hazard index that accounts for the largest component of the study area is mainly distributed up to 0.0010, all mild hazard levels. Guangdong, Guangxi, Hainan, Fujian, Zhejiang, and Jiangsu as well as six other provinces and autonomous regions are all areas with high hazard risks. The research results can provide important scientific evidence for the sustainable development of China's coastal provinces and cities. The outcomes of this study may also provide the scientific basis for the future prevention and mitigation of marine disasters as well as the rationalization of related insurance.

**Keywords:** spatiotemporal pattern; typhoon disaster; kernel density estimation; risk analysis; China

## 1. Introduction

A typhoon is a pattern of severe, disastrous weather. These features are tropical cyclones that occur on the ocean surface in the western North Pacific (i.e., west of the International Date Line, including South China) and include maximum continuous wind forces of more than 12 (wind speed 32.7 m/s) near to their centers [1–3]. China is one of a small number of countries globally that are most influenced by typhoons, and especially in southeast coastal areas. Statistics show that of the ten most significant natural disasters to occur globally between 1947 and 1998, 499,000 people were killed by typhoons, 41% of total deaths from natural disasters [4]. In one example, Typhoon 'Lekima' landed on the coast adjacent to Wenling City (28°22′00.00″ N, 121°20′00.00″ E) in Zhejiang Province on 10 August 2019, and was the third strongest typhoon to land in East China in 70 years. Indeed, as of 13 August 2019 at 16:00 (UTC+8), this typhoon had adversely affected 12,884,000 people across nine provinces (cities), Zhejiang, Shanghai, Jiangsu, Shandong, Anhui, Fujian, Hebei, Liaoning, and Jilin [5]. A total of 2,040,000 people were transferred

urgently and resettled at the time, of which 1,417,000 have now returned. The typhoon also caused the collapse of 13,000 houses and damaged 119,000 to some extent, while crops were affected across an area of 9,960,000 hectares [6].

China is significant affected by typhoons globally due to its close proximity to an active storm zone in the Northwest Pacific. Typhoon disasters are the most frequent and destructive meteorological disasters to occur in China; these are characterized by high frequency, range of impact, strong and sudden activity, significant mass occurrence, and high disaster intensity [7,8]. The dominant hazards associated with typhoon disasters include gale, heavy rain, and storm surges. Rainstorms caused by typhoon, in particular, are extremely destructive. In addition to direct economic losses and casualties, typhoons can also cause secondary disasters such as mountain torrents, mudslides, and collapses. Extreme precipitation records across many parts of China are caused by typhoons; on the islands of Taiwan and Hainan, typhoon-related annual average precipitation versus climatic precipitation can reach up to around 30% [9,10].

Natural disaster risk assessment refers to the appraisal and estimation of hazard factor intensity as well as the degree of potential damage caused by these events determined via risk analysis or the observation appearance method [11–14]. Risk analysis technology can also be applied to natural disasters [15–20].

Current research around the world describes the spatiotemporal characteristics of tropical cyclones in the Northwest Pacific, including inter-annual and seasonal changes in tropical cyclone frequency, typhoon sources, and the intensity distribution and path types of these events landing in China. Significant progress has been made in terms of typhoon disaster risk assessment in recent years [21–23]. The previous typhoon research mainly focused on the typhoon numerical forecast technology [24,25], vulnerability assessment [26,27], post-disaster loss estimation [28–30], and so on. With the advancing of typhoon research, more attention should be paid to the overall structure of the typhoon disaster system. Disaster system theory currently remains limited to the description of hazard spatiotemporal distributions, while work on typhoon disasters has mainly focused on hazard factors and risk assessment across small areas [31]. A limited number of disaster risk assessment studies across large areas have been conducted from a probability perspective [32,33].

The aims of this paper are to: (1) Utilize the kernel density estimation index as a hazard index and to use the probability of exceeding or reaching return period or exceeding a certain threshold to describe the occurrence probability of hazards; and (2) Utilize probability density, transcendence probability, and annual risk to perform national-scale evaluation mapping and result analysis. These approaches have enabled us to compile a disaster risk evaluation map for typhoon landings across China. The research can provide important scientific evidence for the sustainable development of China's coastal provinces and cities and the formulation of disaster prevention and disaster prevention policies.

## 2. Data and Methodology

### 2.1. Data Sources

The typhoons studied in this research comprised the tropical examples listed in the 'Tropical Cyclone Classification (GB)' [34]. Data were taken from the China Meteorological Administration (CMA) 'Best-Track Tropical Cyclone (TC) Data Set' provided by the China Typhoon Network (www.typhoon.org.cn) [35]. Thus, subsequent to the annual tropical cyclone season each year, according to the collected conventional and unconventional meteorological observation data, the path and intensity data of tropical cyclones each year were compiled to form a CMA tropical cyclone best path data set (Table 1).

The current version of the CMA best-track data set provides the position and intensity of tropical cyclones across the Northwest Pacific (including the South China Sea, north of the equator, and west of 180° E) since 1949. These data included typhoon name and number, typhoon center location (latitude and longitude), lowest central pressure (hPa) and two-minute average near-center maximum and average wind speed (m/s).

| Database | Contents | Data Sources | Data Period |
|---|---|---|---|
| CMA TC database | Year, typhoon number, English name, date of creation, end date, minimum central pressure, maximum wind speed, landing longitude and latitude, landing location, affected area, maximum wind force, maximum wind speed, and extreme wind value | Shanghai Typhoon Institute, CMA http://tcdata.typhoon.org.cn/zjljsjj_sm.html | 1949–2018 |
| Typhoon disaster database | Year, typhoon number, wind power, starting time, affected population, number killed, emergency resettlement population, affected area, disaster area, number of collapsed houses, and direct economic losses. | Ministry of Emergency Management of the People's Republic of China | 2000–2018 |

*2.2. Methods*

2.2.1. Typhoon Hazard Index-Kernel Density Estimation (KDE)

The KDE approach provides one non-parametric way to estimate the probability density function of a random variable [36,37]. Thus, KDE is a fundamental data smoothing problem where inferences about population are made based on a finite sample. One well-known KDE application has been for estimating the class-conditional marginal densities of data when using a naive Bayes classifier [38]. This approach dramatically improves prediction accuracy.

We defined $(x_1, x_2, \ldots, x_n)$ a univariate independent and identically distributed sample drawn from some distribution with an unknown density *f*. We are interested in estimating the shape of this function *f*, as follows:

$$\hat{f}_h(x) = \frac{1}{nh}\sum_{i=1}^{n}K(\frac{x - X_i}{h}) \tag{1}$$

$$K(x) \geq 0, \int_{-\infty}^{+\infty}K(x)dx = 1 \tag{2}$$

In this expression, *K* denotes the kernel, a non-negative function, while $h > 0$ is the bandwidth smoothing parameter. A kernel with subscript *h* is referred to as the scaled kernel. Intuitively, it is necessary to define *h* as small as the data will allow and there is always a trade-off between the bias of the estimator and variance. A range of kernel functions are commonly used including uniform, triangular, biweight, triweight, Epanechnikov, and normal. The Epanechnikov kernel is optimal in a mean square error sense even though efficiency loss remains small in the other types [39]. As it has convenient mathematical properties, the normal kernel is often used; this means that $K(x) = \phi(x)$, where $\phi$ is the standard normal density function.

2.2.2. Cumulative and Exceeding Probability, and Return Period

A probability density function describes the relative likelihood that a random variable will take on a given value. The probability of a random variable falling within a particular range is given by the integral of density over that range; in other words, the area under the density function but above the horizontal axis and between the lowest and greatest values [40]. Cumulative frequency analysis is therefore the analysis of value occurrence frequency of a phenomenon less than a reference value [41]. Cumulative frequency is also referred to as the frequency of non-exceedance probability.

A return period, also known as a recurrence interval, provides an estimate of event likelihood [42]. This is a statistical measurement typically based on historic data which denotes the average recurrence interval over an extended period of time and is usually used for risk analysis. Thus, on the basis of conversions from exceeding probability to a certain year return period, typhoon hazard risk maps for once every two years, five years, ten years, and 20 years were obtained.

The specific algorithm used in this study involved determining the exact number of observations in each interval from the frequency histogram and then dividing the number of observations in each interval by the total number of observations. The resultant graph is the relative frequency histogram, plotted at time ordinates between 0 and 1. We then used the sequence $u_1, u_2, \ldots, u_m$ to represent the midpoint of each interval in the histogram, the standard interval point. The relative frequency was then divided by interval length and the result is recorded as i = 1, 2, ... , m. Points were then plotted as i = 1, 2, ... , m; these were connected to get an estimate of the overall probability density function p(x).

Over a certain period of time [0, t], the probability that typhoon disaster intensity is equal to, or exceeds, S is defined as P, and so the cumulative probability function is CF(S). Over the defined time period, the relationship between the probability of exceeding as well as cumulative probability is as follows:

$$P(S) = \frac{CF(S)}{T} \tag{3}$$

We therefore used P(S, t) to represent the probability that hazard intensity is equal to, or exceeds, S within a period of time [0, t]. It therefore follows:

$$P(S, t) = 1 - \left[1 - \left(\frac{CF(S)}{T}\right)\right]^t = 1 - (1 - P(S))^t \tag{4}$$

In one example, suppose that the cumulative number of typhoon disasters exceeding 0.5 over a 100-year period (T = 100) reaches ten, then CF(S) = 10 (Table 2). Thus, over this time period, P(0.5) = 0.1; this means that the probability of one typhoon disaster intensity exceeds 0.5 is 0.1 and the probability of occurrence within one year is 10%. Thus, if we study a typhoon with a disaster intensity greater than 0.5, this formula can be used to obtain the probability of typhoon occurrence at this intensity within a 100-year period. In other words, if t = 10 years, then p(0.5) = 0.1 and p(0.5,10) = 0.75; this means that the probability of a typhoon disaster greater than 0.5 within 10 years is 75%. A return period corresponds to an extreme quantile which represents the numerical magnitude of extreme variables of extreme events. Over a given return period, the larger the extreme quantile, the smaller the probability of exceeding this level as well as the smaller the likelihood of extreme events. The deeper color of the risk maps is represented, the higher relative risk is. The flow of risk analysis of Chinese typhoon disasters is shown in Figure 1.

**Table 2.** Disaster occurrence, cumulative, and exceeding probability over certain return period times.

| Return Period | Exceeding Probability | Cumulative Probability | Occurrence Probability |
|---|---|---|---|
| 2 | 0.50 | 0.50 | 0.50 |
| 5 | 0.20 | 0.80 | 0.30 |
| 10 | 0.10 | 0.90 | 0.10 |
| 25 | 0.04 | 0.96 | 0.06 |
| 50 | 0.02 | 0.98 | 0.02 |
| 100 | 0.01 | 0.99 | 0.01 |
| | 0 | 1.00 | 0.01 |

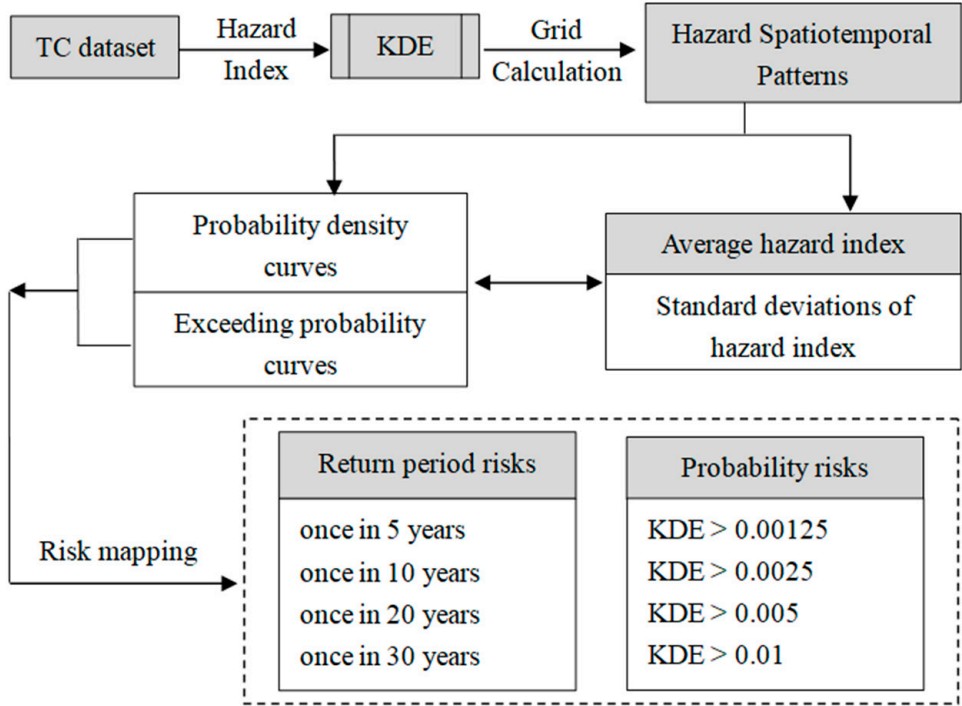

**Figure 1.** Flow chart for risk analysis of Chinese typhoon disasters.

### 3. Results

*3.1. Typhoon Hazard Spatiotemporal Patterns*

The KDE index of 70-year typhoon hazard factor was selected as a sample while 0.005 was determined as the interval length of each histogram. A histogram was then estimated for each evaluation unit to enable us to calculate probability and exceeding probabilities. Larger values of the search radius parameters produce a smoother, more generalized density raster. Smaller values produce a raster that show more detail. The output cell size can be defined by a numeric value or obtained from an existing rater database. By adjusting the parameters and comparing the analysis results, the final search radius parameters is selected as 100km, and the cell size is set as 1km. Thus, for display purposes, based on the annual hazards data samples, annual average KDE indexes for six provinces (i.e., Guangxi, Hainan, Guangdong, Fujian, Zhejiang, and Jiangsu) were calculated. The probability density curves for these hazards were calculated (Figure 2) alongside exceeding probability curves (Figure 3). The probability density curve can include any probability value on the hazard index interval; this enabled us to estimate the probability of different hazard levels. The exceeding probability refers to the chance that the intensity of a typhoon exceeds the intensity of a given event within a certain area and time range. This also reflects the range of hazard risk as well as its changes. Thus, the steeper the probability density curve, the probability of hazard index is greater. The probability of occurrence of this factor is large while the slowly changing section represents a small hazard probability event.

As seen in Figure 2, the hazard index for Jiangsu Province is mainly distributed between 0 and 0.005, while these values for Fujian, Guangdong, and Zhejiang provinces are mainly distributed between 0 and 0.007. The hazard index of Hainan Province is relatively high, mainly between 0.005 and 0.015.

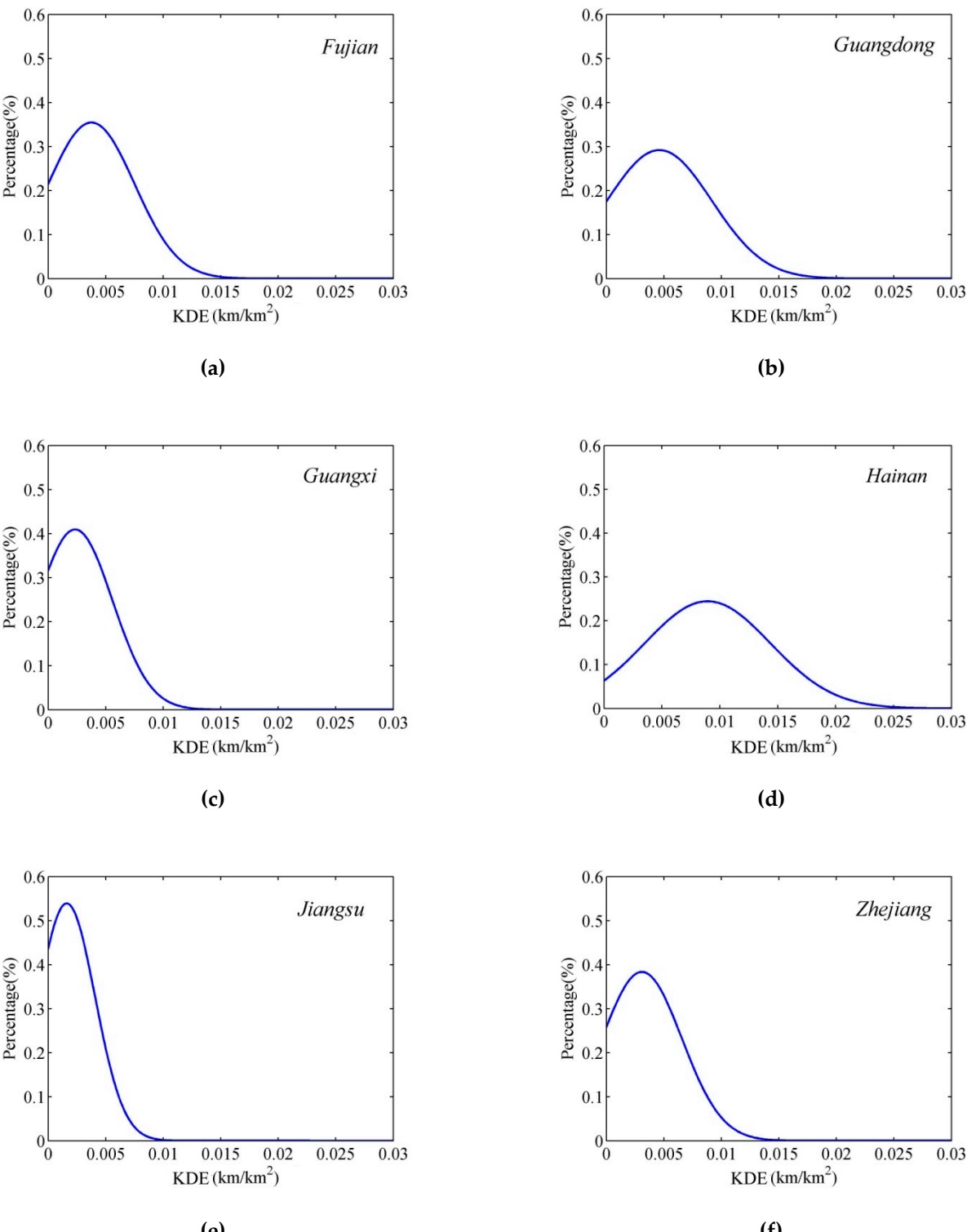

**Figure 2.** Probability density curves for typhoon hazard index across typical areas of China: (**a**) Fujian Province; (**b**) Guangdong Province; (**c**) Guangxi Province; (**d**) Hainan Province; (**e**) Jiangsu Province; (**f**) Zhejiang Province.

The data presented in Figure 3 reveal that the steepest changes in hazard indexes are seen in Fujian, Guangdong, Guangxi, Hainan, Jiangsu, and Zhejiang; 0.01, 0.01, 0.008, 0.015, 0.005, and 0.008, respectively. These points correspond to maximum values on the probability density curve as well as to the enhanced probability of hazard events.

Across the six provinces surveyed here, Hainan is at highest risk. The exceeding probability of the hazard index greater than 0.015 is relatively large in this region, 0.15,

followed by Guangdong Province, where the exceeding probability of the hazard index greater than 0.01 is 0.1. Values for exceeding probability greater than 0.005 in other regions are 0.38 (Fujian Province), 0.3 (Zhejiang Province), 0.2 (Guangxi Province), and 0.08 (Jiangsu Province).

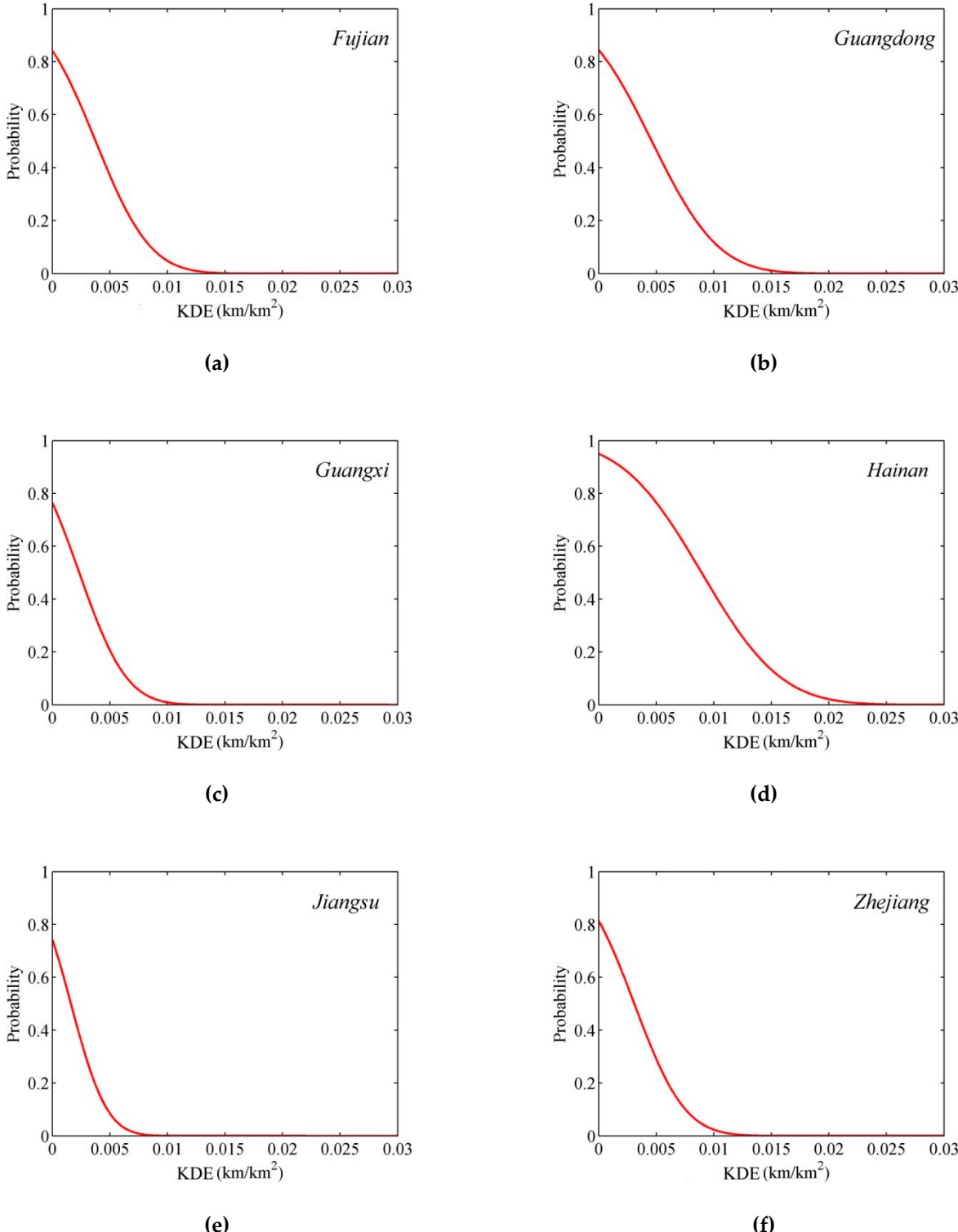

**Figure 3.** The exceeding probability curves of typhoon hazard index across typical areas of China: (**a**) Fujian Province; (**b**) Guangdong Province; (**c**) Guangxi Province; (**d**) Hainan Province; (**e**) Jiangsu Province; (**f**) Zhejiang Province.

We implemented the calculation formula for hazard index; thus, an annual typhoon hazard index value for each grid cell was calculated. Data show the spatial distribution of average hazard index across China between 1949 and 2018 (Figure 4). The spatial distribution of typhoon landing density follows a northeast-southwest zonal distribution across China, decreasing from the southeast coast to the northwest.

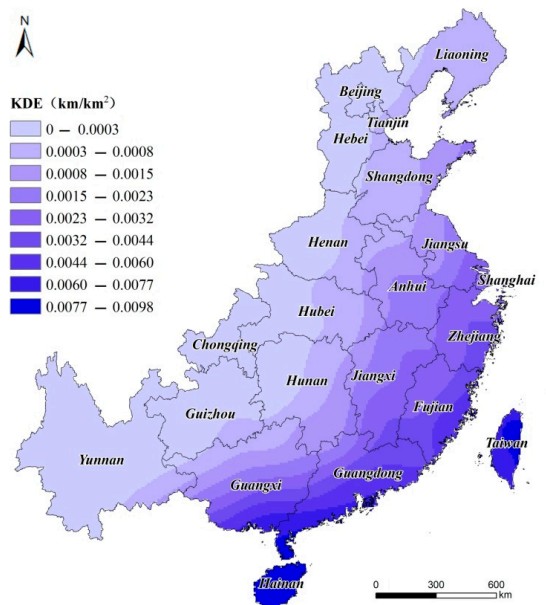

**Figure 4.** Average typhoon hazard index distribution across China.

Coastal areas are the main regions influenced by typhoons and extreme precipitation, most severely in Chinese history. Areas with high hazard index are concentrated on Hainan and Taiwan islands. Regions on the mainland that are severely affected include the coastal areas of Guangdong, Guangxi, and Fujian as well as inland into the middle and lower reaches of the Yangtze River and the middle reaches of the Yellow River. In addition, most areas in Northeast China have also been affected by typhoons. The average values for most hazard indexes are at a low level (0–0.0015), accounting for 67.81% of the total coastal area of China. This area of hazard indexes (0–0.0003) accounts for about 37.51% of total coastal area.

Areas with greater risk typhoon hazard indexes (0.0015–0.0098) are concentrated in the Pearl River and Yangtze River deltas, regions which account for about 32.19% of total coastal area. The high frequencies of typhoons as well as the high level of socioeconomic development across these two regions are the main factors leading to high typhoon disaster risk. The ability to prevent and reduce disaster risk has increased due to improvements in the social economy while the level of typhoon disaster risk remains relatively high due high social wealth concentration. In addition to these two high-risk areas, Shandong, Jiangsu, and other places also have relatively high typhoon risks, followed by the Bohai Rim region.

In order to better reveal fluctuations in the Chinese typhoon hazard index over time and on the basis of typhoon hazard index data for each grid cell, standard deviation was calculated. A standard deviation distribution map for the Chinese typhoon hazard index was then drawn (Figure 5). This map shows that the distribution of standard deviations as a whole reveals obvious differences between east and west with the former larger than the latter. The southern part of Guangdong Province as well as Guangxi Province and Taiwan have the highest standard deviations; in these regions, standard deviations are more than 0.0047 and typhoon hazard indexes exhibits the largest inter-annual fluctuations. The proportion of hazard index standard deviation up to 0.0003 is 20.16%, while the proportion of hazard index standard deviation between 0.0030 and 0.0061 is 23.32%.

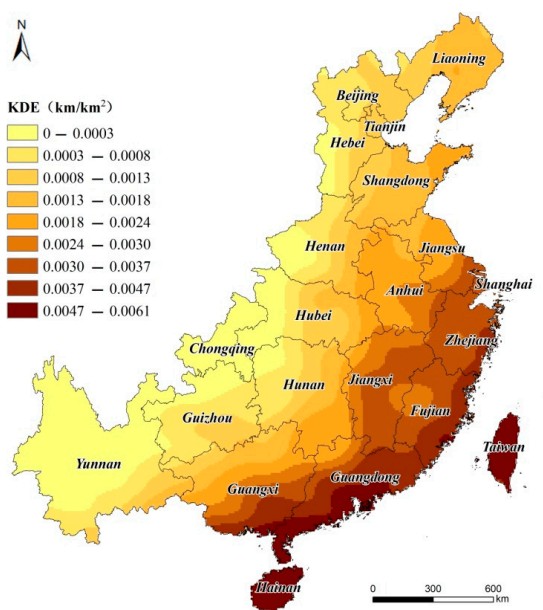

**Figure 5.** Standard deviations of typhoon hazard index distribution across China.

The annual frequency of tropical cyclones generated in the Northwest Pacific including those affecting China has decreased significantly over time. The number of tropical cyclones landing in China has gradually decreased (−0.4 per 10 years), while the frequency of typhoons generated in the Northwest Pacific and the South China Sea has also decreased over time. The frequency of typhoons landing in China has varied greatly year-on-year and the proportion of landed typhoons is increasing. The average number of typhoons that landed in China in the 1990s with wind forces level 12 or higher was 2.8; since 2000, this number has been 4.1, an increase of 46% (Figure 6). From the end of the 1960s to the beginning of the 1970s, the number of landing typhoons was mostly above the multi-year average, and then fluctuated around the multi-year average. From the early 1970s to the mid-1990s, there was a relatively stable fluctuation state. Since the end of the 1990s, there has been a weak downward trend.

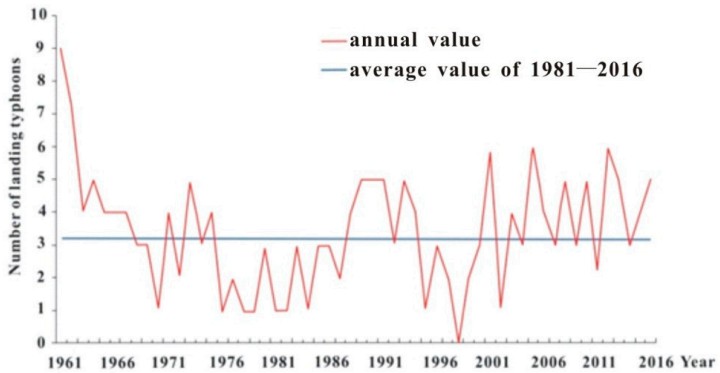

**Figure 6.** Inter-annual changes in the number of typhoons landing in China.

### 3.2. Typhoon Hazard Index Based on Fixed Exceeding Probability

The typhoon hazard index database as well as fixed degrees of exceeding probabilities was used to develop four typhoon hazard risk maps at different risk levels (Figure 7). These results show that 91.52% of maize-planting areas across China fall within the light drought hazard index range between 0 and 0.0010 and correspond with a risk level of one event every two years.

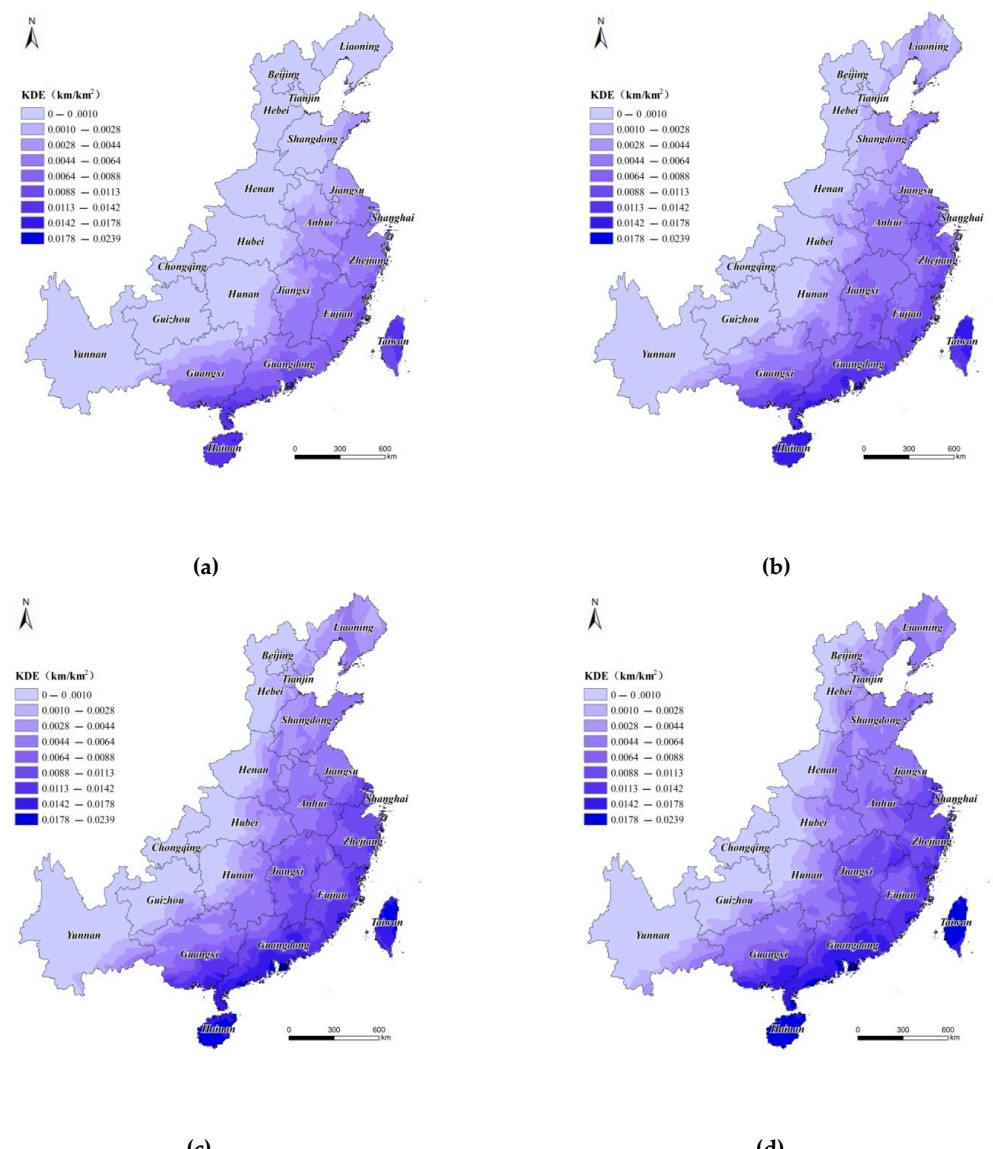

**Figure 7.** Maps showing typhoon hazard index for China at different timescales: (**a**) once in five years, (**b**) once in ten years, (**c**) once in 20 years, and (**d**) once in 30 years.

On the basis of four risk levels, the hazard index that accounts for the largest area is mainly distributed within a range up to 0.0010, all light hazard levels. Thus, given levels of once in five years, once in ten years, once in 20 years, and once in 30 years, the proportions of regions with light hazard levels (up to 0.0010) across the study area were 57.40%, 40.41%, 31.74%, and 27.42%. Irrespective of risk level, Guangxi and southern Guangdong, Hainan, and Taiwan all possess highest hazard intensities, reaching relatively severe levels; hazard index ranges in these cases fall between 0.0178 and 0.0239. Analyses show that, as risk levels increase, the proportion of extreme hazards (0.0113–0.0239) across the study area also gradually increase from 2.73% once in five years, 4.31% once in ten years, 10.87% once every 20 years, up to 13.51% once in 30 years. As risk levels increase, the proportion of light-to-medium hazard levels (0–0.0028) in total areas have increased from 66.16% once in five years, 53.18% once in ten years, 40.56% once in 20 years, to 35.54% once in 30 years.

The distribution of high-value areas, including high-value areas of the hazard index once in five years, are concentrated in Guangdong, Guangxi, and Hainan provinces. These values are determined by the fact that the southeast coastal area is the one mainly affected by typhoon extreme precipitation. The hazard index value in parts of central and northern

parts of China remain low, below the light hazard level. As risk levels increase, 10-year, 20-year, and 30-year once-in-a-time levels feed into unchanged risk patterns while high-value areas increase significantly. High-value areas of the hazard index gradually expand to the northeast and northwest.

### 3.3. Probability Risk Based on Fixed Typhoon Hazard

Another type of hazard risk map was generated to calculate the probability of occurrences under different hazard indexes using the exceeding probability of each grid cell. This was then used to draw a corresponding series of risk maps, including four hazard index levels, typhoon hazard index > 0.00125 (Figure 8a), typhoon hazard index > 0.0025 (Figure 8b), typhoon hazard index > 0.0005 (Figure 8c), and typhoon hazard index > 0.01 (Figure 8d).

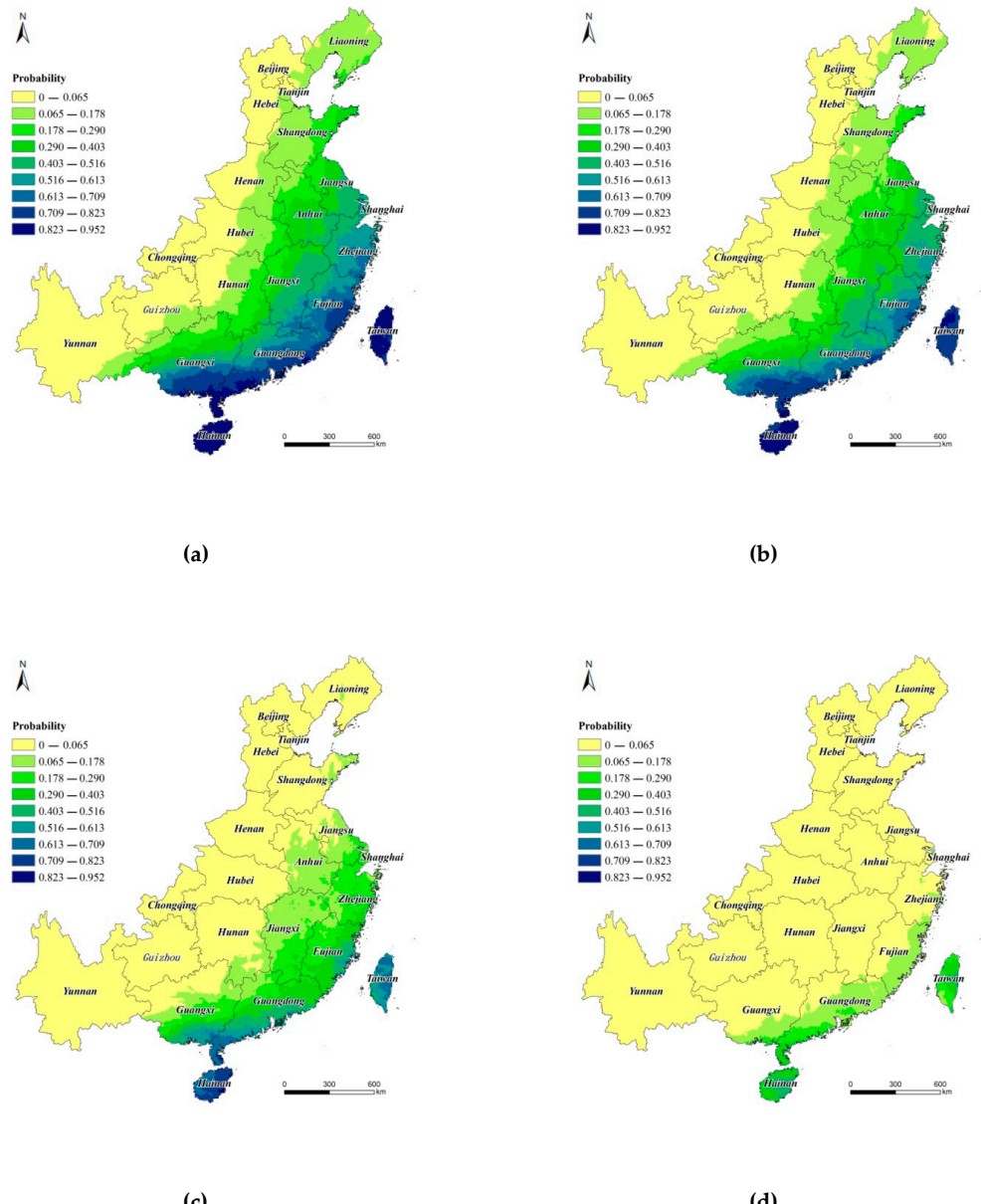

**Figure 8.** Maps showing probability risks, given different typhoon hazard indexes across China: (**a**) Kernel Density Estimation (KDE) > 0.00125; (**b**) KDE > 0.0025; (**c**) KDE > 0.005; and (**d**) KDE > 0.01.

In light of different hazard index levels, data show that the southeast coastal area of China is the area with highest typhoon occurrence probability. Indeed, amongst these areas, Guangdong, Guangxi, Hainan, Fujian, Zhejiang, and Jiangsu provinces have the darkest

colors. The coastal areas of China are not only densely populated but the economies in these areas are most developed. Social wealth is high, but because these areas are close to the Pacific Ocean, they are directly affected by typhoons and can therefore become high risk regions.

## 4. Discussion

This research was initiated using typhoon track statistical data over many years. We used the influence mechanisms of typhoon frequency and landing latitudes while our knowledge regarding the main factors affecting typhoons currently remains unclear. A typhoon is often accompanied by strong winds, heavy rains and high tide events [43,44]. Multivariate return period analysis can provide more adequate and comprehensive information about risks than univariate return period analysis. The prevention measures such as critical infrastructures design, should consider the multiple hazard factors of typhoon disasters comprehensively, and give full consideration to the occurrence probability during the design period. The principle of extreme value statistics was used to estimate the main meteorological disaster factors of typhoons and the extreme value distribution probability types of disaster indicators [45,46]. Probability information regarding hazard factors and disaster loss information were linked to construct a disaster risk model. We strengthened the application of typhoon risk assessment demonstration area and then used this to determine whether, or not, a disaster is caused. The magnitude of a disaster risk, how big this risk is (disaster risk index), and whether a catastrophe occurs (the disaster risk index and the probability of exceeding disaster loss based on smaller low frequency characteristics) should be analyzed in the future.

As the foundation of the disaster risk prevention, disaster assessment is composed of loss estimate and risk analysis [18,42]. The vulnerability analysis is an important component in disaster assessment, connecting analysis of hazard and risk [47–49]. As the quantitative and accurate assessment means of vulnerability, the vulnerability curve has been widely used in disaster estimate, quantitative risk analysis and risk mapping. Some scholars have given a deep research to the mechanism of typhoon-flood disaster chain and produce a house damage assessment model of five southeast coastal provinces using comprehensive disaster magnitude as parameter [4]. With the rapid development of modern information technology and the in-depth study of disasters, theories that can reflect the nature of vulnerability—the vulnerability curve will become the future development trend.

Due to the numerous and complex factors affecting the typhoon disaster risk, it is difficult to analyze the risk completely objectively and quantitatively. The typhoon hazard risk model based on the KDE index is only a preliminary exploration. When making typhoon disaster risk zoning, more factors should be considered, such as people's awareness of disaster prevention and mitigation and disaster mitigation facilities. This will be further studied in the future. A new method should also be developed to analyze regional typhoon hazards. The frequency of typhoon influence should also be considered alongside intensity and duration [50–52]. The comprehensive effect of typhoons on each affected area can also be quantified more comprehensively and objectively. These can better reflect the regional differentiation of typhoon hazards and enable more accurate regional typhoon disaster risk assessment.

The disaster system theory should be used to analyze the risk of typhoons from four aspects: hazard factors, disaster-affected body, disaster-inducing environment, and disaster prevention and mitigation capacities [15–17]. For practical use, emergency management interventions should be intensified from three aspects: hazard mitigation measures, emergency preparedness measures, and recovery measures to minimize the impact of disasters. Hazard mitigation measures refer to fully inspecting the intensity, scope, and continuity of hazard factors before a disaster, and raising public awareness through emergency knowledge publicity and education, and preventing and controlling the severity of disasters. Emergency preparedness measures refer to the systematic preparations for disaster prevention and mitigation before a disaster, including preparations by governments, enterprises,

the masses, and non-governmental organizations. Recovery measures refer to the adoption of active measures during the post-disaster recovery period to make the disaster social state return to normal as soon as possible. Therefore, the conclusions could only give reference opinions. The physical mechanism can be further explored in the following research to better analyze the typhoon risks.

Coastal cities are highly vulnerable to typhoons due to the special geographical location. Once affected by a typhoon, it will cause greater economic losses in coastal areas of China, especially the impact on agriculture and fishery production. Under the influence of a super typhoon, industry and tertiary industries may also be forced to be interrupted, which will affect the entire city. Conversely, human activities may have a certain impact on the intensity and path of a typhoon [3,4]. For example, the density of urban construction and vegetation coverage affect the friction of the ground during typhoon activities. The construction of major projects has profoundly changed the landscape of the city and will also have a corresponding impact on the typhoon. The next step in this research will be to analyze the causal relationships of typhoon events. Catastrophic losses and extremes in typhoon hazards, the absence of disaster loss, and the normality of hazards also need to be assessed. In term of mitigating hazards, we should strengthen the accuracy of typhoon forecast, get through the "last one kilometer" of emergency management, reinforce professional training and drills of emergency management personnel, and enhance the masses emergency publicity and education.

## 5. Conclusions

We used the CMA TC database to select the southeastern coastal area of East China as our research area. The data for various typhoon elements between 1949 and 2018 (a total of 70 years) was statistically processed and the paths that landed in China were assessed. We used the KDE as the hazard index; thus, the probability of exceeding, or reaching, return period or exceeding a certain threshold was used to describe the probability of hazard occurrence.

The results of this analysis reveal that the overall spatial distribution of typhoon hazards conforms to a northeast-southwest zonal distribution, decreasing from the southeast coast to the northwest. Thus, across the six typical provinces of China assessed here, data show that Hainan possesses the highest hazard risk. Hazard index is relatively high, mainly distributed between 0.005 and 0.015, while the probability of exceeding a hazard index greater than 0.015 is 0.15.

In light of the four risk levels assessed here, the hazard index that accounts for the largest component of the study area is mainly distributed up to 0.0010, all mild hazard levels. Guangdong, Guangxi, Hainan, Fujian, Zhejiang, and Jiangsu as well as six other provinces and autonomous regions are all areas with high hazard risks. The coastal areas of China are not only densely populated, but their economies are most developed; social wealth is high, these regions are close to the Pacific Ocean and are directly affected by typhoons, and they are all high hazard areas.

**Author Contributions:** Conceptualization, F.C. and H.J.; methodology, H.J. and L.W.; software, N.W.; validation, E.D. and N.W.; formal analysis, A.Y.; investigation, L.W.; resources, F.C. and H.J.; data curation, N.W.; writing—original draft preparation, H.J.; writing—review and editing, F.C.; supervision, F.C.; project administration, F.C.; funding acquisition, H.J. All authors have read and agreed to the published version of the manuscript.

**Funding:** This research was funded by the International Partnership Program of the Chinese Academy of Sciences (131211KYSB20170046), the National Natural Science Foundation of China (41671505; 41871345), and the National Key R&D Program of China (2017YFE0100800).

**Institutional Review Board Statement:** "Not applicable" for studies not involving humans or animals.

**Informed Consent Statement:** "Not applicable" for studies not involving humans.



**Data Availability Statement:** No new data were created or analyzed in this study. Data sharing is not applicable to this article.

**Acknowledgments:** This work was supported by the International Partnership Program of the Chinese Academy of Sciences (131211KYSB20170046), the National Natural Science Foundation of China (41671505; 41871345), and the National Key R&D Program of China (2017YFE0100800). The authors also would like to appreciate the editors and the three anonymous reviewers for their constructive comments and advice.

**Conflicts of Interest:** The authors declare no conflict of interest.

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
