# Peer review of "Spatiotemporal Variations and Risk Analysis of Chinese Typhoon Disasters"

_sustainability, doi:10.3390/su13042278_

Round 1

Reviewer 1 Report

The revised manuscript appears to be more comprehensive and authors adopted reviewers' comments and it is in a mature stage for publications.

Author Response

The revised manuscript appears to be more comprehensive and authors adopted reviewers' comments and it is in a mature stage for publications.

Reply:

Thank you for your affirmation and encouragement. We appreciate for Reviewers’ warm review work earnestly. The comments are all valuable and very helpful for revising and improving our paper. Once again, thank you very much for your comments and suggestions.

Reviewer 2 Report

The author obviously did not make any meaningful changes.

What the reviewer cares about is that this is an article about typhoon disasters, but there is no basic typhoon path and rainfall data?

Kernel Density Estimation (KDE) is only a mathematical estimation method.

The author can use it, but it is not the core result.

Readers will not feel any interest in KDE.

If the author quantified the typhoon path and rainfall data first, it would be acceptable to discuss KDE.

Author Response

The author obviously did not make any meaningful changes. What the reviewer cares about is that this is an article about typhoon disasters, but there is no basic typhoon path and rainfall data?

Reply:

We greatly appreciate the reviewers' comment. The reviewer may have ignored the data and methods of our manuscript. The key data of this manuscript is the typhoon path data; please refer to 2.1 Data Sources. (Page 2, line 87-94; Page 3, line 95-100).

Kernel Density Estimation (KDE) is only a mathematical estimation method. The author can use it, but it is not the core result. Readers will not feel any interest in KDE. If the author quantified the typhoon path and rainfall data first, it would be acceptable to discuss KDE.

Reply:

Thank you for pointing this out. In the authors' opinion, the biggest issue in the current study of typhoons is that the risk is difficult to quantify. Typhoon is not just a line, it is on the surface. In the authors' opinion, the key innovation of the manuscript is how to solve the surface problem. The KDE provides a feasible technical support for this issue in this paper. Kernel density estimation is one way to convert a set of points (an instance of vector data) into a raster. Of course, Kernel Density Estimation (KDE) is NOT only a mathematical estimation method. KDE is also a method of spatial analysis in ArcGIS software (refer to the following picture).

The path data of the typhoon can provide the basis for calculating the frequency of occurrence. The current risk analysis of this manuscript is done from this perspective. Of course, the typhoon has too many factors (including intensity and duration), and the characteristics of the city should also be taken into consideration. These are also reflected in the Data sources and Discussion part of this manuscript. (Page 2, line 87-94; Page 16, line 311-324; Page 17, line 325-375; Page 18, line 376-377).

Reviewer 3 Report

This is an interesting study on typhoon distribution analysis. The authors used KDE to determine the risk levels of each region/zone. Overall I find it a nice read. No major concerns. However, I don't see new findings from the conclusions. As we all know from common sense, southeast coast is prone to typhoon hazard. I would suggest authors to put some more discussions on how the results would be useful. And here are some comments:

  1. Your discussion/conclusion is mostly on large scale, but for practical use, how would one city apply these findings to design defense structures?
  2. In Figure 5, the larger numbers in 1961-1963 don't mean the trend is decreasing, right? It is more towards flat after 1965. I'm just wondering how this figure can help you with your conclusion.

Author Response

This is an interesting study on typhoon distribution analysis. The authors used KDE to determine the risk levels of each region/zone. Overall I find it a nice read. No major concerns. However, I don't see new findings from the conclusions. As we all know from common sense, southeast coast is prone to typhoon hazard. I would suggest authors to put some more discussions on how the results would be useful.

Reply:

Thank you for pointing this out. Special thanks to you for your good comments. It is really true as reviewer suggested that some more discussions should be put. We have made modifications according to the Reviewer’s comments (Page 16, line 314-320; Page 17, line 340-345; Page 17, line 351-374).

And here are some comments:

  1. Your discussion/conclusion is mostly on large scale, but for practical use, how would one city apply these findings to design defense structures?

Reply:

We agree with this comment. Thank you for pointing this out. For practical use, disaster reduction capacity construction should be built on the basis of a relatively complete disaster risk analysis. By analyzing the intensity-frequency and spatial differentiation of typhoon disaster risk, combined with the capacity assessment results, the capacity building gap analysis should be carried out. And then the detailed demand analysis should be carried out. Then set regional typhoon disaster reduction targets in a certain stage, and finally implement them into specific defense projects. Emergency management interventions should be intensified from three aspects: hazard factor mitigation measures, emergency preparedness measures, and recovery measures to minimize the impact of disasters. We have modified “Discussions” according to the Reviewer’s comments (Page 16, line 314-320; Page 17, line 340-345; Page 17, line 351-374).

  1. In Figure 5, the larger numbers in 1961-1963 don't mean the trend is decreasing, right? It is more towards flat after 1965. I'm just wondering how this figure can help you with your conclusion.

Reply:

Special thanks to you for your good comments. The average number of typhoons that landed in China in the 1990s with wind forces level 12 or higher was 2.8 (The emphasis here is with wind forces level 12 or higher); since 2000, this number has been 4.1, an increase of 46%. From the end of the 1960s to the beginning of the 1970s, the number of landing typhoons was mostly above the multi-year average, and then fluctuated around the multi-year average. From the early 1970s to the mid-1990s, there was a relatively stable fluctuation state. Since the end of the 1990s, there has been a weak downward trend. We have added a detailed explanation of Figure 5 in the revised manuscript (Page 12, line 255-258).

We appreciate for Reviewers’ warm review work earnestly. The comments are all valuable and very helpful for revising and improving our paper, as well as the important guiding significance to our researches. We have studied comments carefully and have made correction which we hope meet with approval. Once again, thank you very much for your comments and suggestions.

Reviewer 4 Report

This study proposes introduces the spatiotemporal pattern analytics for Chinese Typhoon Disasters with kernel density estimation (KDE). It is a very interesting and timely topic. The topic is related to sustainability. More specific comments can be found in the following,

1) The literature review part is not sufficient. The related work part is needed.

2) The detailed workflow will help to understand the whole process of the whole study.

3) Could you please give more descriptions of how to choose the bandwidth for the kernel density estimation (KDE)?

4) All maps should be improved (north arrow should be added).

5) Line 103.( We defined (x1, x2, ..., xn) should be (x1, x2, ..., xn).

6) Is that possible to open the sample data and source code for reproducible research?

Author Response

This study proposes introduces the spatiotemporal pattern analytics for Chinese Typhoon Disasters with kernel density estimation (KDE). It is a very interesting and timely topic. The topic is related to sustainability. More specific comments can be found in the following,

  • The literature review part is not sufficient. The related work part is needed.

Reply:

We agree with this comment. Thank you for pointing this out. We have added the related work according to the Reviewer’s comments (Page 2, line 70-74).

The previous typhoon research mainly focused on the typhoon numerical forecast technology [18, 19], vulnerability assessment [20, 21], post-disaster loss estimation [22–24], and so on. With the advancing of typhoon research, more attention should be paid to the overall structure of the typhoon disaster system.

The following is the added reference list.

  • Li, H.; Luo, J.Y.; and Xu, M.T. Ensemble data assimilation and prediction of typhoon and associated hazards using TEDAPS: evaluation for 2015-2018 seasons. Frontiers of Earth Science. 2019, 13(4), 733-743.
  • Zhang, R.; Zhang, W.J.; Zhang, Y.J.; Feng, J.N.; and Xu, L.T. Application of Lightning Data Assimilation to Numerical Forecast of Super Typhoon Haiyan (2013). Journal of Meteorological Research. 2020, 34(5), 1052-1067.
  • Niu, H.Y.; Liu, M.; Lu, M.; Quan, R.S.; Zhang, L.J.; and Wang, J.J. Risk Assessment of Typhoon Disasters in China Coastal Area During Last 20 Years. Scientia Geographica Sinica. 2011, 31(6), 764-768.
  • Nguyen, K.A.; Liou, Y.A.; and Terry, J.P. Vulnerability of Vietnam to typhoons: A spatial assessment based on hazards, exposure and adaptive capacity. Science of the Total Environment. 2019, 682, 31-46.
  • Chen, C.J.; Lee, T.Y.; Chang, C.M.; and Lee, J.Y. Assessing typhoon damages to Taiwan in the recent decade: return period analysis and loss prediction. Natural Hazards. 2018, 91(2), 759-783.
  • Liu, Y.; Li, N.; Zhang, Z.T.; and Chen, X. Indirect economic loss and its dynamic change assessment of typhoon Ewiniar in Guangdong. Journal of Catastrophology. 2019, 34(3), 178-183.
  • Yu, J.; Zhao, Q.S.; and Chin, C.S. Extracting Typhoon Disaster Information from VGI Based on Machine Learning. Journal of Marine Science and Engineering. 2019, 7(9), 318.

  • The detailed workflow will help to understand the whole process of the whole study.

Reply:

We agree with this comment. The flow of risk analysis of Chinese typhoon disasters is shown in Figure 1. We have added the detailed workflow according to the Reviewer’s comments (Figure 1, Page 5, line 159-161). Thank you for pointing this out.

  • Could you please give more descriptions of how to choose the bandwidth for the kernel density estimation (KDE)?

Reply:

We greatly appreciate the reviewers' comment. In the previous manuscript, “0.005 was determined as the bandwidth” is a writing error. We have checked it carefully. 0.005 was determined as the interval length of each histogram. Larger values of the search radius parameters produce a smoother, more generalized density raster. Smaller values produce a raster that show more detail. The output cell size can be defined by a numeric value or obtained from an existing rater database. By adjusting the parameters and comparing the analysis results, the final search radius parameters is selected as 100km, and the cell size is set as 1km. Then this part has been rewritten in the revised manuscript (Page 5, line 164-171).

  • All maps should be improved (north arrow should be added).

Reply:

We agree with this comment. We have added the north arrow for all the maps in the revised manuscript (Page 10-11; Page 13, line 287-290; Page 15; Page 16, line 307-310).

  • Line 103.( We defined (x1, x2, ..., xn) should be (x1, x2, ..., xn).

Reply:

Thank you for pointing this out. We have modified it according to the Reviewer’s comments (Page 3, line 108).

  • Is that possible to open the sample data and source code for reproducible research?

Reply:

Special thanks to you for your good comments. Of course we would like to open the sample data and source code for reproducible research. This is very beneficial for sharing and improving our research methods. We can then provide a public URL to upload the technical manual to the website for readers' reference. In the future, we plan to share it on the following websites http://data.casearth.cn/. We greatly appreciate the reviewers if you could give us understanding and support. Thanks very much.

Round 2

Reviewer 2 Report

From the main four results in the author's manuscript (Figures 4, 5, 7, and 8), we can see that the coastal typhoon disaster risk is relatively high. Based on the results of the time series shown in Figure 6, the author claims that there have been fewer typhoons recently landed, but did not make any analysis to explore the causes, and also did not analyze which area has fewer typhoons. The author puts forward the most important conclusion in this manuscript based on the KDE analysis results: "The results of this analysis reveal that the overall spatial distribution of typhoon hazards conforms to a northeast-southwest zonal distribution, decreasing from the southeast coast to the northwest.". However, for researchers who study typhoons in the Northwest Pacific, such results do not require KDE analysis to achieve similar results.

The author claims in the manuscript that "A typhoon is often accompanied by strong winds, heavy rains and high tide events.", but the author has not made any substantial data analysis of the above factors.

The author also admits that "The typhoon hazard risk model based on the KDE index is only a preliminary exploration." In addition, most of what the author mentioned in the "Discussion" chapter has no substantive scientific significance, but only estimates the impact of the typhoon.

It is a pity that the author still insists on his own opinions and cannot persuade me. Here, I cannot encourage more.

Author Response

From the main four results in the author's manuscript (Figures 4, 5, 7, and 8), we can see that the coastal typhoon disaster risk is relatively high. Based on the results of the time series shown in Figure 6, the author claims that there have been fewer typhoons recently landed, but did not make any analysis to explore the causes, and also did not analyze which area has fewer typhoons.

Reply:

We greatly appreciate the reviewers' comment. The research on risk in this manuscript is based on the probability risk of typhoon path data in long-term series (1949-2018). The typhoon hazard index database as well as fixed degrees of exceeding probabilities was used to develop four typhoon hazard risk maps. Another type of hazard intensity risk map was generated to calculate the probability of occurrences under different hazard intensity indexes (Page 12, line 263-265; Page 14, line 294-296). Indeed from Figure 6 we can see the number of tropical cyclones landing in China has gradually decreased (-0.4 per 10 years). This only reflects the overall trend. For visualization, the author divides KDE into different levels( for example 0-0.0003; 0.0003-0.0008; 0.0008-0.0015; 0.0015-0.0023; 0.0023-0.0032; 0.0032-0.0044; 0.0044-0.0060; 0.0060-0.0077; 0.0077-0.0098 in Figure 4) according to the histogram distribution of the data. The dark areas in Figures 4, 5, 7, and 8 can only indicate that the relative risk is high. We also have made modifications in the revised manuscript (Page 4, line 155-157). Therefore, the analysis angles of Figure 6 and Figures 4, 5, 7, and 8 are different. The two parts are not comparable.

The author puts forward the most important conclusion in this manuscript based on the KDE analysis results: "The results of this analysis reveal that the overall spatial distribution of typhoon hazards conforms to a northeast-southwest zonal distribution, decreasing from the southeast coast to the northwest.” However, for researchers who study typhoons in the Northwest Pacific, such results do not require KDE analysis to achieve similar results.

Reply:

Thank you for pointing this out. Special thanks to you for your good comments. The overall spatial distribution is a fact that cannot be changed, and it is indeed the result of a series of analyses of this manuscript. The biggest problem in the current study of typhoons is that the risk is difficult to quantify. Typhoon is not just a line, it is on the surface. In the authors' opinion, the key innovation of the manuscript is how to solve the surface problem. The KDE provides a feasible technical support for this issue in this manuscript. The impact frequency and intensity of the landfalling typhoon in China are distributed in a band, gradually decreasing from the coast to the inland. In addition to the overall spatial distribution, there are specific risk quantitative classification and spatial analysis (Page 9, line 217-242; Page 12, line 268-286; Page 14, line 294-305).

The author claims in the manuscript that "A typhoon is often accompanied by strong winds, heavy rains and high tide events.", but the author has not made any substantial data analysis of the above factors.

Reply:

We agree with this comment. Thank you for pointing this out. This conclusion "A typhoon is often accompanied by strong winds, heavy rains and high tide events." is that the authors directly cited published references. It is really true as reviewer suggested that some references should be put. We have added “references” according to the Reviewer’s comments (Page 16, line 317; Page 20, line 516-520).

The author also admits that "The typhoon hazard risk model based on the KDE index is only a preliminary exploration." In addition, most of what the author mentioned in the "Discussion" chapter has no substantive scientific significance, but only estimates the impact of the typhoon.

Reply:

Thank you for pointing this out. Of course, the typhoon has too many factors (including intensity, duration, and degree of aggregation), and the characteristics of the city should also be taken into consideration. Not only just estimates the impacts of the typhoon (Page 17, line 367-377), from the perspective of engineering facilities such as critical infrastructures design (Page 16, line 316-323), vulnerability curves analysis (Page 17, line 332-341), damage assessment model (Page 16, line 324-332), typhoon disaster risk zoning(Page 17, line 344-351), practical use based on the disaster system theory (Page 17, line 353-365), the authors have discussed in detail in the “Discussion” part.

It is a pity that the author still insists on his own opinions and cannot persuade me. Here, I cannot encourage more.

Reply:

We appreciate for Reviewers’ warm review work earnestly. You have encouraged us a lot. The comments are all valuable and very helpful for revising and improving our paper, as well as the important guiding significance to our researches. We have studied comments carefully and have made correction which we hope meet with approval. Once again, thank you very much for your comments and suggestions.

This manuscript is a resubmission of an earlier submission. The following is a list of the peer review reports and author responses from that submission.

Round 1

Reviewer 1 Report

The manuscript presents an analysis of Kernel Density Estimation regarding typhoon landfall over China. Disaster risk evaluation maps for typhoon landings across China are provided illustrated the relevant impact. Authors used statistical tools to analyse and compile probability and risk maps. My suggestion is a Minor Revision taking into account the following general and minor points in order the manuscript to be more consistent.

General Point

As the journal is addressed to international readers, it is very difficult for the readers that are unfamiliarized with the geography of China to follow the results and the discussion on this paper. Authors are using names of provinces and cities that are not illustrated in the figures. I strongly suggested to authors to introduce in all maps the provinces’ name and make relevant connections in the manuscript.

Minor Points

  1. Line 42: Either insert the lat&lon info for the city or illustrate it on a map
  2. Iine 43= Please, use UTC time zone
  3. Line 49: Either insert a reference for that statement or rewrite it as “China is significant affected by …”.
  4. Authors introduced the term Kernel Density Index (line 71), however in the manuscript KDE is presented. I suggest to authors to use it throughout the manuscript.
  5. I suggest to authors to rename Section 2 from "Materials and Methods" to "Data and Methodology"
  6. Fig.7: KD is not defined in the manuscript. Please rewrite the figure caption

Reviewer 2 Report

It is difficult for the reviewer to understand the important scientific purpose and conclusion that the author wants to express?
Modularization is a method.
But the whole article is not innovative enough,
on the other hand,
We only need to draw the path of the typhoon to calculate the density of the passage to achieve the purpose of the author's expression.
This article lacks scientific and social quality,
The typhoon has too many factors, and the characteristics of the city should also be taken into consideration.